# Empowering Multi-Robot Cooperation via Sequential World Models

**Zijie Zhao**[1,2], **Honglei Guo**[2], **Shengqian Chen**[2], **Kaixuan Xu**[1,2], **Bo Jiang**[2,1],
**Yuanheng Zhu**[2,1,*], **Dongbin Zhao**[2,1]
[1]School of Artificial Intelligence, University of Chinese Academy of Sciences
[2]SKL-MAIS, Institute of Automation, Chinese Academy of Sciences

## Abstract

Model-based reinforcement learning (MBRL) has achieved remarkable success in robotics due to its high sample efficiency and planning capability. However, extending MBRL to physical multi-robot cooperation remains challenging due to the complexity of joint dynamics. To address this challenge, we propose the **Seq**uential **W**orld **M**odel (SeqWM), a novel framework that integrates the sequential paradigm into multi-robot MBRL. SeqWM employs independent, autoregressive agent-wise world models to represent joint dynamics, where each agent generates its future trajectory and plans its actions based on the predictions of its predecessors. This design lowers modeling complexity and enables the emergence of advanced cooperative behaviors through explicit intention sharing. Experiments on Bi-DexHands and Multi-Quadruped demonstrate that SeqWM outperforms existing state-of-the-art model-based and model-free baselines in both overall performance and sample efficiency, while exhibiting advanced cooperative behaviors such as predictive adaptation, temporal alignment, and role division. Furthermore, SeqWM has been successfully deployed on physical quadruped robots, validating its effectiveness in real-world multi-robot systems. Demos and code are available at: SeqWM.

## 1 Introduction

Model-based reinforcement learning (MBRL) has been widely applied to robotic systems due to its high sample efficiency (Jiang et al., 2025) and planning capability (Sun et al., 2023). However, extending MBRL to multi-robot cooperation remains challenging (Chai et al., 2025). Early decentralized approaches built independent world models for each agent (Egorov & Shpilman, 2022), overlooking coupling between agents and hindering coordination. More recent centralized methods, by contrast, assume full observability (Liu et al., 2024b), performing dynamics modeling (Zhao

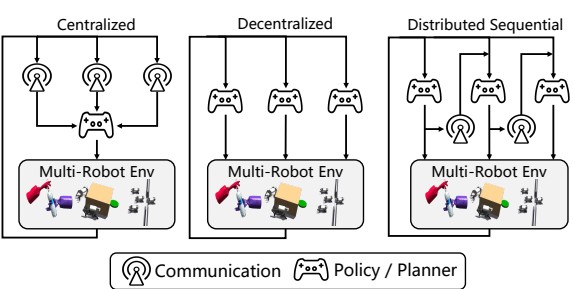

Figure 1: Comparison of SeqWM's distributed sequential paradigm with existing centralized/decentralized paradigms.

et al., 2025a) and policy optimization (Zhang et al., 2025a) in the joint space. These methods face challenges related to modeling complexity in robotic systems with high-dimensional observation and action spaces, limiting their deployment in real-world scenarios.

Between centralized and decentralized paradigms, the distributed sequential paradigm has rapidly developed in recent years and demonstrated unique advantages (Khan, 2025). It reformulates multi-agent decision-making as an autoregressive process: agents communicate and act in a certain order, with each updating its policy conditioned on messages and actions from predecessors (Wen et al.,

*Corresponding authors. This work was supported in part by National Natural Science Foundation of China under Grants 62136008, 62293541, and in part by Beijing Nova Program under Grant 20240484514.

2022; Hu et al., 2025). This design enables more consistent joint reasoning (Ding et al., 2024) and finer-grained credit assignment (Kuba et al., 2022) without relying on full communication. From a real-world deployment perspective, it reduces the reliance on communication synchronization and offers improved robustness against packet loss or disturbances (Ding et al., 2024).

Motivated by these advantages, as shown in Figure 1, we propose the **Seq**uential **W**orld **M**odel (SeqWM), which integrates the sequential paradigm into MBRL to structurally decompose dynamics modeling and action planning. For trajectory prediction, SeqWM represents the joint dynamics as sequential agent-wise rollouts, where each agent maintains an independent world model and conditions on the predicted trajectories and actions of its predecessors. For action planning, each agent performs multi-step lookahead conditioned on its predecessors' predictions, thereby preserving cooperative performance while constraining the search to a low-dimensional subspace consistent with the sequential structure. We evaluate SeqWM in two challenging multi-robot cooperation environments: Bi-DexHands (Chen et al., 2024a) and Multi-Quad (Xiong et al., 2024), and further validate its effectiveness on physical multi-robot tasks using two Unitree Go2-W robots. The key contributions are as follows:

(1) By integrating the sequential paradigm, SeqWM decomposes joint dynamics into autoregressive agent-wise models, reducing modeling complexity, thereby extending MBRL to multi-robot cooperation.

(2) Through explicit intention sharing, SeqWM enables the emergence of advanced cooperative behaviors such as predictive adaptation, temporal alignment, and role division.

(3) SeqWM consistently outperforms all baseline methods on the simulated multi-robot benchmarks and demonstrates successful real-world deployment on a physical multi-quadruped platform.

These results collectively demonstrate that SeqWM, by leveraging sequential world modeling and planning, offers an effective pathway for multi-robot cooperation, balancing performance, efficiency, and real-world applicability.

## 2 RELATED WORK

**Model-based RL.** In robotics, model-based RL has shown remarkable success due to its high sample efficiency (Jiang et al., 2025), with several approaches (Sun et al., 2023) leveraging learned dynamics models to predict trajectories and optimize actions for physical robots. In contrast, existing multi-robot MBRL methods often rely on centralized paradigms (Zhao et al., 2025b), hindering their practical deployment in multi-robot systems. For example, CoDreamer (Toledo, 2024) use transformers or GNNs to integrate full state-action across all agents. Recent efforts such as MARIE (Zhang et al., 2025a) explored decentralized dynamics modeling, but still require communication at each prediction step for agent-wise aggregation. Different from these works, SeqWM assigns each agent an independent world model and predicts trajectories sequentially, which structurally lowers modeling complexity, thereby making it applicable to real-world scenarios.

**Sequential Paradigm.** Recent studies have highlighted the advantages of the sequential paradigm (Khan, 2025), which enables fine-grained credit assignment (Kuba et al., 2022), efficient dynamics modeling (Zhang et al., 2025b), and scalable coordination (Xu et al., 2025). For example, MAT (Wen et al., 2022) and PMAT (Hu et al., 2025) model the multi-agent decision-making process as a sequence prediction problem, employing transformers to autoregressively predict each agent's actions. HARL (Liu et al., 2024a; Zhong et al., 2024) further introduce the sequential update scheme that bring clearer interpretability and ensure monotonic improvement. SeqComm (Ding et al., 2024) extends this idea to the communication domain, where agents exchange information in a sequential order, effectively mitigating non-stationarity. Motivated by these benefits, we integrate the sequential paradigm into multi-robot world modelling to enhance planning and coordination.

## 3 PRELIMINARIES

**Problem Formulation.** We model the fully cooperative task as a decentralized partially observable Markov decision process (dec-POMDP), $\mathcal{M} = \langle \mathcal{I}, \mathcal{S}, \mathcal{O}, \Omega, \mathcal{P}, \mathcal{R}, \gamma \rangle$, where $\mathcal{I} = \{v^1, \dots, v^n\}$ is the set of agents, $\mathcal{S}$ is the global state space, $\mathcal{O} = \prod_{i=1}^{n} O^i$ is the joint observation space, and

$\mathcal{A} = \prod_{i=1}^{n} A^i$ is the joint action space. The observation function $\Omega : \mathcal{S} \times \mathcal{I} \to \mathcal{O}$ defines each agent's perception of the environment, while the transition function $\mathcal{P} : \mathcal{S} \times \mathcal{A} \to \mathcal{S}$ specifies the environment dynamics. The reward function $\mathcal{R} : \mathcal{S} \times \mathcal{A} \to \mathbb{R}$ provides a shared scalar signal, and $\gamma$ is the discount factor. Each agent $v^i$ learns a local policy $\pi^i : O^i \to A^i$, which maps its observation $o^i$ to an action $a^i$. The objective is to learn a joint policy $\pi = \prod_{i=1}^{n} \pi^i$ that maximizes the expected discounted return $\sum_{\tau=t}^{\infty} \gamma^\tau r_\tau$.

**Sequential Decision-Making.** In many real-world applications, multi-robot systems are often distributed rather than fully decentralized (Negenborn & Maestre, 2014), allowing inter-agent communication to enhance cooperative performance. A Dec-POMDP can thus be extended to a multi-agent POMDP (Oliehoek et al., 2016), where each agent $v^i$ receives messages $e_t^i$ from other agents and updates its policy $\pi^i : O^i \times E \to A^i$. To balance efficiency and decision quality, agents adopt communication protocols, defined as $\phi^i : O^i \times E \times A^i \to E$. Among them, sequential protocols is especially popular for its simplicity and effectiveness (Ding et al., 2024). It organizes agents in a certain order, where each agent acts on its own observation and the message from its predecessor, then passes information forward. Formally, the process is defined as:

$$a_t^i = \pi^i(o_t^i, e_t^i), \quad e_t^{i+1} = \phi^i(e_t^i, o_t^i, a_t^i), \tag{1}$$

Such a sequential structure naturally motivates us to design a world model that predicts trajectories in the same manner, enabling efficient multi-agent planning.

## 4 METHODOLOGY

In this section, we propose **SeqWM**, which decomposes the joint dynamics into agent-wise models arranged in a sequence. This design substantially reduces modeling complexity, enabling deployment in physical multi-robot systems.

### 4.1 SEQUENTIAL WORLD MODELLING

**Decomposed Joint Dynamics.** At each timestep $t$, the observation-action pair of a single agent $(o_t^i, a_t^i)$ can be regarded as a token, and the entire system as a sequence of such tokens. This perspective reformulates joint dynamics as a sequence modeling problem: given the token sequence $[(o_t^1, a_t^1), \ldots, (o_t^n, a_t^n)]$, the dynamics generates the next-step outcomes $[(o_{t+1}^1, r_{t+1}^1), \ldots, (o_{t+1}^n, r_{t+1}^n)]$. Unlike existing centralized world models, which fuse all tokens simultaneously for prediction, as shown in Figure 1, our method adopts an autoregressive paradigm. In this setup, agent 1 first predicts $(o_{t+1}^1, r_{t+1}^1)$ from its local information $(o_t^1, a_t^1)$, and passes the result to 2. Subsequently, each agent $i$ conditions on its own observation–action pair $(o_t^i, a_t^i)$ together with the predictions of all predecessors $\{(o_{t+1}^j, r_{t+1}^j)\}_{j<i}$ to produce $(o_{t+1}^i, r_{t+1}^i)$. Such a sequential design reduces modeling complexity in a structured, scalable manner, making the approach well-suited for real-world deployment.

**World Model.** As noted in INTRODUCTION, multi-robot cooperative tasks involve high-dimensional observation and action spaces, making it unsuitable to use reconstructing raw observations as the learning objective of the world model (Hansen et al., 2022). Therefore we remove the explicit decoder and instead perform dynamics prediction entirely in a latent space. To facilitate distributed deployment, each agent maintains an independent world model without parameter sharing. Let $z_t^i$ denote the latent state of agent $v^i$ at timestep $t$, the world model can be defined as follows:

$$
\begin{aligned}
\text{Encoder:} \quad z_t^i &= E^i(o_t^i) \\
\text{Dynamics:} \quad \hat{z}_{t+1}^i &= D^i(z_t^i, a_t^i, e_t^i) \\
\text{Reward:} \quad \hat{r}_{t+1}^i &= R^i(z_t^i, a_t^i, e_t^i) \\
\text{Communication:} \quad e_t^{i+1} &= e_t^i \oplus a_t^i \\
\text{Critic:} \quad \hat{q}_t^i &= Q^i(z_t^i, a_t^i, e_t^i) \\
\text{Actor:} \quad \hat{a}_t^i &= \pi^{i,\text{Act}}(z_t^i, e_t^i).
\end{aligned}
\tag{2}
$$

All modules in SeqWM are implemented using MLPs, ensuring architectural simplicity and consistency. For communication function, we adopt concatenation operator $\oplus$ to facilitate modular training. As shown in Section 5.5, this concise design achieves more stable training performance compared to alternatives such as cross-attention and recurrent neural networks (RNNs).

**Learning Objective.** Let $\theta_E$, $\theta_D$, $\theta_R$, $\theta_Q$, and $\psi$ denote the parameters of the encoder, dynamics predictor, reward predictor, critic, and actor, respectively. Following the self-supervised training framework (Hansen et al., 2024), the loss functions can be defined as:

$$\mathcal{L}^i(\theta) = \sum_t^H \lambda^t \left( \underbrace{\left\| \hat{z}^i_{t+1} - \text{sg}(z^i_{t+1}) \right\|^2}_{\text{dynamics loss for } \theta_D, \theta_E} + \underbrace{\text{Soft-CE}\left(\hat{r}^i_t, r_t\right)}_{\text{reward loss for } \theta_R, \theta_E} + \underbrace{\text{Soft-CE}\left(\hat{q}^i_t, G_t\right)}_{\text{Q Loss for } \theta_Q, \theta_E} \right), \tag{3}$$

where $\theta = \{\theta_E, \theta_D, \theta_R, \theta_Q\}$, $H$ is the prediction horizon, $\lambda \in (0,1]$ is a constant that balances the contribution of each rollout step, $r_t$ is the ground-truth reward, $G_t$ is TD target, and $\hat{z}^i_{t+1} = D^i\left(z^i_t, a^i_t, e^i_t\right)$ is the predicted latent state. The loss can be backpropagated to the encoder via $z^i_t = E^i\left(o^i_t\right)$, so do the dynamics and reward losses. The latent target $z^i_{t+1} = E^i\left(o^i_{t+1}\right)$ is detached with the stop-gradient operator $\text{sg}(\cdot)$ to prevent cyclic gradient flow. Soft Cross-Entropy loss is used to match the discretized reward and Q-value predictions. Additionally, to ensure modularity and scalability, each agent's world model is trained independently, and the loss can not be backpropagated through the communication channel.

Based on Eq. (3), the encoder learns a compact latent space, while the dynamics and reward predictors minimize prediction errors in this space, ensuring alignment with real environment dynamics. The actor generates initial action estimates in the latent space to warm-start planning and is trained using the Heterogeneous-Agent Soft Actor-Critic (HASAC) (Liu et al., 2024a) algorithm:

$$\mathcal{L}(\psi) = \sum_t^H \lambda^t \left( Q^i\left(z^i_t, \hat{a}^i_t, e^i_t\right) - \alpha \mathcal{H}\left[\pi^{i,\text{Act}}\left(\cdot \,\middle|\, z^i_t, e^i_t\right)\right] \right), \tag{4}$$

where $\alpha$ is the entropy coefficient, and $\mathcal{H}[\cdot]$ denotes the entropy function.

We adopt the sequential update scheme (Kuba et al., 2022; Zhong et al., 2024) to train world models in a manner aligned with its autoregressive structure. When training the agent $v^{i+1}$, its inputs are conditioned on the predictions of the first $i$ agents, produced by their most recently updated models. This preserves the sequential dependency, ensuring predictions exploit the most up-to-date outputs, which stabilizes training and improves monotonicity across agent indices.

Inspired by Masked AutoEncoders (He et al., 2022), we randomly permute the order among agents and allow each agent to skip communication with a certain probability. This random masking simulates realistic interruptions and forces the world model to robustly adapt to uncertainties, significantly enhancing the model's resilience against communication failures.

## 4.2 SEQUENTIAL PLANNING

Although Eq. (2) includes an actor, it does not serve directly as an explicit decision policy; instead, it provides initial action estimates for the planner. We next propose a sequential multi-agent planner based on Model Predictive Path Integral (MPPI) (Williams et al., 2015) that leverages the predictions of world models to optimize each agent's action. In this framework, the actor contributes by narrowing the action search space to promising regions, while the planner ensures robust long-term decision-making and corrects suboptimal proposals from the actor.

At each timestep $t$, agent $v^i$ samples $N$ candidate action sequences of horizon $H$, denoted $a^i_{t:t+H}$, from the initial distribution guided by the actor. Conditioned on its latent state and the received message, the agent performs latent rollouts with its local world model to predict future

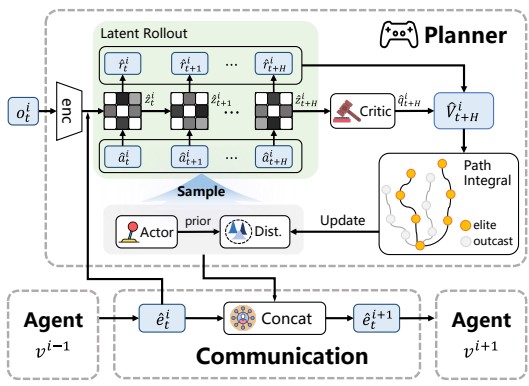

Figure 2: Sequential planner: agents sequentially optimize actions via local world models and share planned trajectories.

trajectories. The value of each trajectory is then estimated as

$$V_{t+H}^i = \gamma^H Q^i(\hat{z}_{t+H}^i, a_{t+H}^i, e_{t+H}^i) + \sum_{h=t}^{t+H-1} \gamma^{h-t} R^i(\hat{z}_h^i, a_h^i, e_h^i), \qquad (5)$$

where $V_{t+H}^i$ represents the value estimate of a sampled action sequence, computed as the sum of predicted rewards over the horizon plus the terminal value given by the critic. Then, candidate sequences are ranked according to their evaluated values, and the highest-scoring subset is selected as the elite set. The action distribution is updated toward the statistics of these elite trajectories, thereby concentrating future sampling around high-value regions and progressively refining the search space. For further details, please refer to Appendix A.1.

After several iterations until convergence, the optimized action sequence and predicted trajectory are transmitted as a message to the next agent, which repeats the same planning procedure. This sequential planning paradigm substantially enhances multi-agent cooperation efficiency through explicit intention sharing.

**Low-pass Action Smoothing.** To prevent mechanical wear caused by abrupt action changes, we integrate a low-pass filtering strategy (Kicki, 2025). In each planning iteration, sampled action sequences are filtered along the temporal dimension to suppress high-frequency fluctuations. This smoothing enforces gradual action transitions across timesteps, reducing control discontinuities and promoting stable, consistent behavior on physical robots. Further details are provided in Appendix A.2.

**Heuristic Early-Stopping.** Considering the computational constraints of physical robotic platforms, we design a motion-planning heuristic that terminates iterations when the KL divergence between consecutive action distributions falls below a threshold. This early-stopping criterion mitigates diminishing returns (Kobilarov, 2012), reducing computation while preserving plan quality. Further details and experiments are provided in Appendix C.2.

**Communication Cache.** Inspired by action-chunking (Li et al.) which reuses temporally extended action units to improve decision efficiency, we introduce a cache that stores the predicted messages from the previous agent, enabling the current agent to retrieve them when communication fails. For instance, if communication fails at $t + 1$, agent $v^{i+1}$ retrieves the cached message $z_{t+1}^i = D^i(E^i(o_t^i))$ from agent $v^i$ instead of the ideally updated message $\hat{z}_{t+1}^i = E^i(o_{t+1}^i)$.

## 5 EXPERIMENTS

**Environments.** We evaluate SeqWM and baselines in two challenging multi-robot cooperative environments: Bimanual Dexterous Hands (Bi-DexHands) (Chen et al., 2024a) and Multi-Quadruped Environment (Multi-Quad) (Xiong et al., 2024). In Bi-DexHands, two agents control a pair of dexterous hands to accomplish high-dimensional manipulation tasks (up to $\mathcal{O} \in \mathbb{R}^{229}, \mathcal{A} \in \mathbb{R}^{26}$). In Multi-Quad, multiple quadruped robots collaborate to solve coordination tasks, and we further deploy SeqWM on real Unitree Go2-W robots to assess its sim-to-real transfer.

### 5.1 COMPARISONS

**Baselines.** We select several competitive baselines, including: HASAC (Liu et al., 2024a), a state-of-the-art model-free method extending SAC to multi-agent settings; MARIE (Zhang et al., 2025a), a model-based method employing a Transformer for dynamics prediction; MAT (Wen et al., 2022), a method adopting the sequential decision-making paradigm; and MAPPO (Yu et al., 2022), a most widely used algorithm, included as a general-purpose baseline.

**Results on Bi-DexHands.** The representative tasks in Bi-DexHands include object transfer tasks (`Over`, `CatchAbreast`, `CatchOver2Underarm`), which require the two hands to transfer an object under different relative positions and grasping postures; and functional manipulation tasks (`BottleCap`, `Pen`, `Scissors`), which involve precise bimanual operations to achieve specific functional goals, such as opening a bottle cap, removing a pen lid, or spreading a pair of scissors.

As shown in Figure 3, SeqWM achieves higher asymptotic returns and faster convergence across all tasks. In several tasks (`Over`, `CatchOver2Underarm`, `Scissors`), SeqWM reaches near-

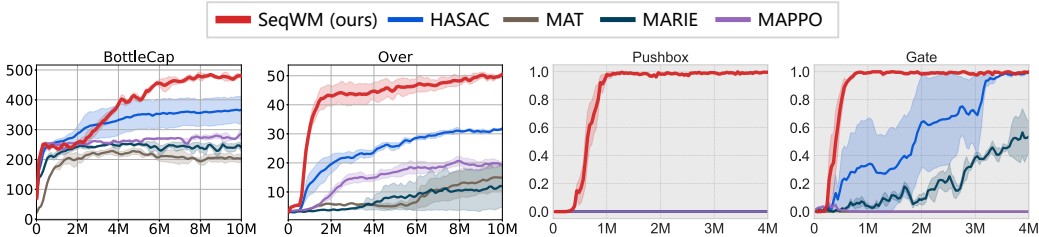

Figure 3: Performance comparisons on selected tasks of SeqWM with other baselines. Task in Bi-DexHands report the episode return, while Multi-Quad (gray background) reports success rate. Bold lines indicate the mean over multiple seeds, with shaded regions denoting the 95% confidence intervals. The results on all other tasks are reported in Figure 12 in Appendix C.1.

optimal performance within 2–4M steps, while baselines require far more interactions or fail to match it. In more challenging tasks (`Pen`, `CatchAbreast`), SeqWM steadily improves and achieves the highest final returns with lower variance, demonstrating stability.

**Results on Multi-Quad.** In `Gate`, the robots are required to pass through a narrow gate as quickly as possible without collision. In `PushBox`, they jointly push a large box to a designated target location. In `Shepherd`,, the two quadruped robots (as sheepdogs) cooperatively guide another robot (as sheep) to a target area (as sheep pen).

In `Gate` and `Shepherd`, it rapidly approaches near-100% success rates within the early phase, significantly surpassing baselines in terms of sample efficiency. This superior performance stems from SeqWM's sequential structure, which enables each agent to plan actions conditioned on its predecessors' intentions, thereby enhancing coordination.

## 5.2 EMPOWERED COOPERATIVE BEHAVIORS

We further visualize the behaviors learned by SeqWM, showing that it not only acquires stable policies in high-dimensional state and action spaces, but also achieves advanced cooperative behaviors, including **predictive adaptation**, **temporal alignment**, and **role division**.

**Bi-DexHands Behaviors.** In `Catch-Over2Underarm`, the throwing hand first performs prediction and planning, explicitly transmitting future trajectories to the catching hand. Guided by this message, the catching hand then exhibits **predictive adaptation** by anticipating the object's motion and landing point and proactively adjusting its grasping posture. As shown in Frames C–D, the catching hand lowers and opens in advance, aligning its posture with the predicted landing point to enable a reliable grasp. In `Pen`, two hands achieve near-perfect **temporal alignment** by exchanging predictions of future actions in advance. As a result, they grasp the pen body and cap almost simultaneously in Frame D and efficiently complete the extraction in Frames E–F, substantially enhancing cooperative efficiency.

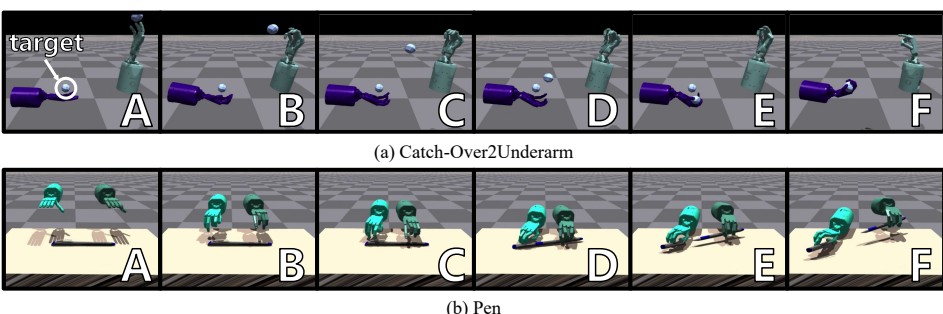

(a) Catch-Over2Underarm

(b) Pen

Figure 4: Trajectory visualizations of `Catch-Over2Underarm` and `Pen` with SeqWM.

**Multi-Quad Behaviors.** Figure 5 further shows the role division learned by SeqWM in the Multi-Quad-`PushBox`. In Frames A–B ($t = 1 \rightarrow 2$), the two quadruped robots navigate to opposite sides

of the box, establishing an effective pushing configuration. In Frames B–D ($t = 2 \rightarrow 4$),, both maintain high positive $x$-axis velocities, indicating continuous forward pushing force. At Frame C ($t = 3$), Robot 2 produces a downward $y$-axis velocity, adjusting the push direction toward the target, while Robot 1 gradually increases its negative $y$-axis velocity to assist in directional control. As the box approaches the target, Robot 1 reduces its $x$-axis velocity to avoid overshooting. These behaviors demonstrate that SeqWM supports not only effective force coordination but also fine-grained directional adjustments, resulting in precise and efficient task completion.

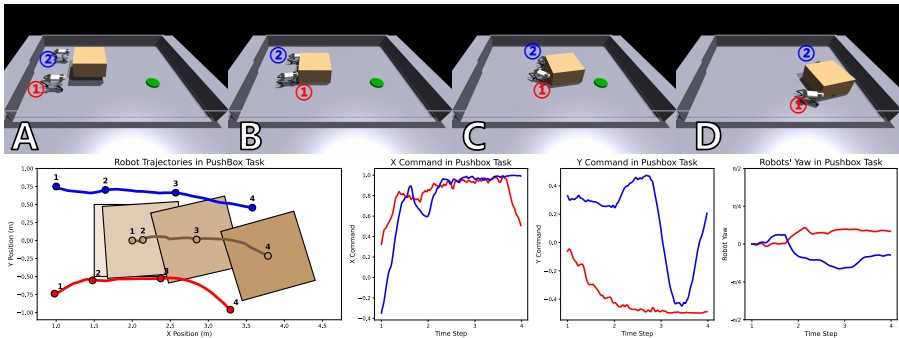

Figure 5: Behavior visualizations in `PushBox`. The first row shows the execution process, where the box is significantly larger than the robots, requiring coordinated efforts from both quadrupeds to complete the task. The left side of second row visualizes the trajectories of the robots and the box, with the right side showing the x-axis and y-axis velocities and orientations of each robot.

## 5.3 SCALABILITY TO MORE AGENTS

We extend the `Gate` to 5 agents to evaluate the scalability of SeqWM, and the behavioral visualizations of `5-robot-Gate` are presented in Figure 6.

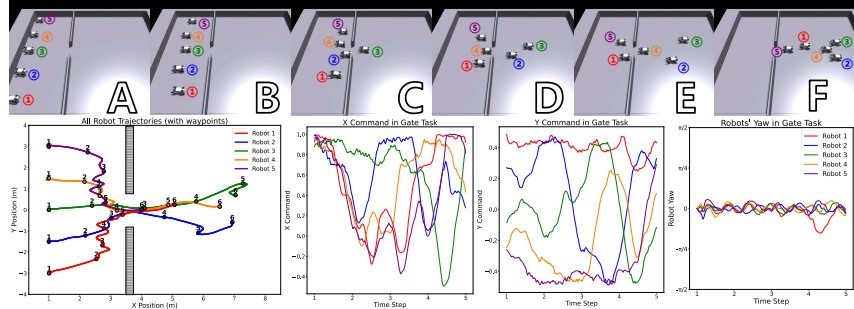

Figure 6: Visualization of the learned behaviors on `5-robot-Gate`.

As the robots approach the narrow gate, they exhibit predictive adaptation, with certain agents proactively decelerating or waiting to avoid potential congestion. For instance, at Frame B ($t\approx2$), Robot 3 maintains a near-unity positive x-command, while the other robots moderately reduce their forward commands. In terms of temporal alignment, the x-command trajectories reveal a clear wave-like alternation, where the peak sequence ($3 \rightarrow 2 \rightarrow 1 \rightarrow 4 \rightarrow 5$) mirrors the actual passing order, reflecting a dynamic "first-pass–then-follow" sequence. Overall, the team establishes a coordinated rhythm of "prediction–waiting–passing–yielding," which enables efficient multi-robot traversal under constrained environmental conditions.

## 5.4 REAL-WORLD DEPLOYMENT

The real-world experimental setup is detailed in Appendix C.3, and the results are shown in Figure 7.

In `PushBox`, the two quadrupeds approach the box from opposite sides and coordinate their pushing forces and directions to move it toward the target. Between Frames D-F, Robot 1 moves forward

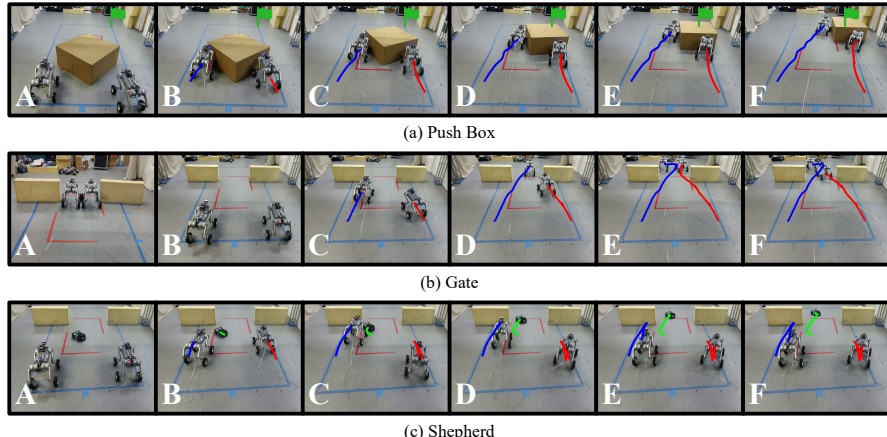

(a) Push Box

(b) Gate

(c) Shepherd

Figure 7: Real-world results of multi-robot cooperation tasks. The trajectories of Robot 1, Robot 2, and the Sheep are marked in different colors.

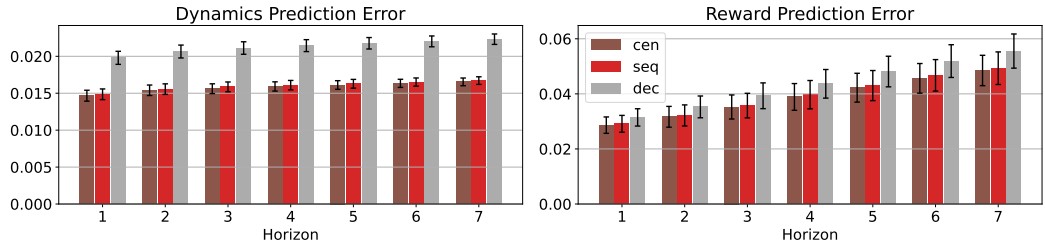

Figure 8: Dynamics (left) and reward (right) prediction errors across horizons.

to provide the main pushing force, while Robot 2 makes slight lateral adjustments to steer the box. The overall pushing pattern, including the division of roles and the gradual directional adjustments, closely matches the behavior observed in simulation, confirming a successful sim-to-real transfer.

In Gate, two clear yielding events are observed. Between Frames C-D, Robot 1 slows down and waits for Robot 2 to pass first, demonstrating priority management in constrained spaces. After crossing Frames E-F, Robot 2 actively veers aside to leave sufficient space for Robot 1, enabling smooth passage without collisions. These behaviors reflect SeqWM's trajectory prediction and intention-sharing capabilities, allowing natural, efficient yielding.

In Shepherd, Robot 1 accelerates between Frames A-B, causing the Sheep to move left. To prevent the Sheep from hitting the left gate frame, Robot 1 retreats while Robot 2 advances between Frames C-D. This maneuver drives the Sheep away from Robot 2 and into the target area. The sequence highlights SeqWM's capacity for predictive coordination and adaptive role allocation, where the one agent's motion influences the sheep robot's response and the another agent adapts accordingly to achieve the common goal.

## 5.5 ABLATION STUDIES

**Sequential Sample Generation.** To evaluate the contribution of the sequential paradigm in SeqWM's world model, we replace it with centralized and decentralized architectures, ensuring all models have an equal number of parameters for a fair comparison. Using BottleCap, we collect 50K environment steps with random actions and train each model for 2.5K steps using the loss in Eq. (3). After training, we gather 1K additional steps to measure dynamics and reward prediction errors across different horizons. As shown in Figure 8, the sequential and centralized models achieve similarly low errors, both substantially outperforming the decentralized model. The results confirm the advantage of sequential prediction, where each agent conditions its output on the predictions of its predecessors, yielding more accurate and coherent rollouts.

**Communication Function.** We replaced the concat communication function in SeqWM with alternative fusion mechanisms, including MLP, cross-attn, and RNN, and evaluated them on `BottleCap`. The results in Figure 9 (left) show that the simplest concat approach achieves the highest and most stable performance. This advantage stems from two factors: (i) concat preserves the complete communication content, allowing the dynamics and reward predictors to autonomously identify and exploit the most informative features during training; and (ii) it introduces no additional learnable parameters, thereby maintaining stable gradient propagation in long-horizon prediction. Moreover, we observe that RNN-based fusion even underperforms the no-communication baseline (dec), which we attribute to its sensitivity to input ordering—an undesirable property in multi-agent communication scenarios lacking a fixed semantic sequence.

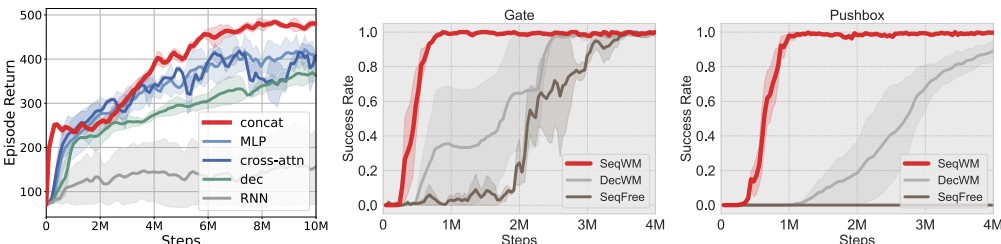

Figure 9: *(Left)*: Performance ablation of the communication functions. *(Middle & Right)*: Ablation of sequential intention sharing.

**Sequential Intention Sharing.** A key insight behind SeqWM is that cooperation becomes substantially easier when an agent can access its partners' future plans. To validate this mechanism, we construct two ablation groups. *DecWM* removes sequential intention sharing and uses a decentralized world model without inter-agent trajectory passing, while *SeqFree* removes the world model entirely and allows agents to exchange only single-step messages.

As shown in Figure 9, SeqWM consistently achieves the best performance, followed by DecWM, whereas SeqFree performs the worst. This indicates that both the world model for predicting future trajectories (i.e., intentions) and the sequential communication for sharing these intentions are indispensable. The gap between SeqWM and DecWM shows that explicit multi-step intention sharing is crucial: without receiving communicated future trajectories from others, each agent can only plan based on its own rollout, which makes coordinated patterns harder to acquire. At the same time, DecWM still clearly outperforms SeqFree, since the decentralized world model implicitly learns regularities in how other agents tend to respond when predicting future environment evolution under joint actions, enabling more foresighted planning even without explicit intention exchange.

## 6 CONCLUSION

This paper presented SeqWM, a novel framework that integrates the sequential paradigm into world model learning and planning. By structurally decomposing joint dynamics into autoregressive, agent-wise models, SeqWM offers a principled approach that reduces modeling complexity and naturally enables intention sharing through predicted trajectories. This methodological innovation not only improves scalability but also facilitates the emergence of advanced cooperative behaviors such as predictive adaptation, temporal alignment, and role division. Extensive experiments in Bi-DexHands and Multi-Quad show that SeqWM achieves state-of-the-art performance with superior sample efficiency, while real-world deployment on quadruped robots confirms that these cooperative behaviors transfer reliably from simulation to physical platforms. Beyond empirical results, SeqWM demonstrates that sequential paradigms provide an efficient and scalable principle for structuring multi-agent cooperation, paving the way for more robust and efficient deployment of cooperation in physical multi-robot systems.

**Limitations.**

SeqWM is designed for fully cooperative tasks with shared rewards, and its effectiveness in competitive or mixed-motive settings remains unexplored. Moreover, the current framework relies on a

fixed or random order, lacking the ability to dynamically adjust the agent sequence during execution, which may limit performance in scenarios where role priorities change over time.

**Future Work.** Benefiting from the integration of the sequential paradigm and agent-wise world models, SeqWM naturally extends to heterogeneous robot teams and human–robot semantic understanding. With each agent maintaining an independent world model, the framework accommodates diverse dynamics and sensing modalities, enabling cooperation among quadrupeds, manipulators, and aerial robots. Moreover, the explicit trajectory rollouts can be shared not only across robots but also with humans as interpretable intention signals, fostering transparent collaboration, mutual understanding, and trust in human–robot teams.

## ETHICS STATEMENT

This work focuses on advancing multi-robot cooperation through model-based reinforcement learning. All experiments were conducted in simulated environments and on standard quadruped robot platforms, with no involvement of humans, animals, or sensitive personal data. The proposed methodology focuses on technical contributions to the fields of reinforcement learning, multi-agent systems, and robotics, without ethical implications beyond standard academic research practices.

## REPRODUCIBILITY STATEMENT

Hyperparameters and training procedures are detailed in Appendix B. All baseline comparisons use publicly available implementations, with the documented parameter settings as referenced in the respective sections.

## USE OF LARGE LANGUAGE MODELS

Large Language Models (LLMs) were used in the preparation of this paper exclusively for writing assistance and language polishing. The conceptualization of the research, methodology design, experimental implementation, and analysis were all conducted entirely by the authors. The authors take full responsibility for the content of this paper.

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

APPENDICES

# A   MULTI-AGENT PLANNER

## A.1   PLANNING PROCESS

At timestep $t$, the action planning process for agent $v^i$ can be divided into the following steps:

**S1 - Communication.** Agents are organized to exchange messages in a sequential manner. Specifically, agent $v^i$ receives a message $e_t^i$ that aggregates the predicted latent states and planned actions from all its predecessors:

$$e_t^i = \begin{cases} \emptyset, & i = 1, \\ \bigoplus_{j<i} \left( \hat{z}_t^j, a_t^j \right), & i > 1, \end{cases} \tag{6}$$

where $\oplus$ denotes concatenation. To implement this efficiently, we employ a masking-based concatenation scheme: a fixed-length vector of dimension $n \times (|\mathcal{A}| + d_z)$ is pre-allocated, where $n$ is the number of agents, $|\mathcal{A}|$ and $d_z$ are the action and latent dimensions. Agent $v^1$ maintains an empty message, while subsequent agents sequentially fill in their designated slots with their own predictions $(\hat{z}_t^i, a_t^i)$ in addition to forwarding the received content. This design ensures that information is progressively accumulated along the communication chain with linear complexity in the number of agents.

**S2 - Action Sampling.** The planner samples $N$ candidate action sequences from two sources. We sample $N_p$ candidate action sequences from a diagonal Gaussian distribution $a_{t:t+H}^i \sim \mathcal{N}\left( \mu_{t:t+H}^i, (\sigma_{t:t+H}^i)^2 I \right)$, where $\mu_{t:t+H}^i, \sigma_{t:t+H}^i \in \mathbb{R}^{|\mathcal{A}| \times H}$ represent the mean and standard deviation of the $H$-step horizon actions. Additionally, we sample $N_a$ action sequences directly from the actor module $\hat{a}_h^i \sim \pi^{i,\text{Act}}(\cdot | o_h^i, e_h^i), h = t : t + H$, and combine these two sets of action sequences to form $N$ candidate action sequences.

**S3 - World Model Prediction.** Following sampling, the world model predicts $H$-step trajectories for each sampled action sequence using Eq. (2), generating $N$ predicted sequences $\Gamma = \{(\hat{z}_h^i, a_h^i, \hat{r}_h^i)\}_{h=t:t+H}$.

**S4 - Value Evaluation.** Each predicted trajectory is assigned a value via the $H$-step return, combining the short-term cumulative predicted reward with the terminal value from the critic:

$$V_\Gamma^i = \gamma^H Q^i(\hat{z}_{t+H}^i, a_{t+H}^i, e_{t+H}^i) + \sum_{h=t}^{t+H-1} \gamma^{h-t} \hat{r}_h^i. \tag{7}$$

**S5 - Action Optimization.** The candidate action sequences are ranked by their evaluated values, and the top $M$ are chosen as the elite set $\Gamma^*$. The parameters of the action distribution are updated based on the elite set using:

$$\mu_{t:t+H}^{i,(k+1)} = \frac{\sum_{m=1}^M \alpha_m \Gamma_m^*}{\sum_{m=1}^M \alpha_m}, \quad \sigma_{t:t+H}^{i,(k+1)} = \sqrt{\frac{\sum_{m=1}^M \alpha_m \left( \Gamma_m^* - \mu_{t:t+H}^{i,(k+1)} \right)^2}{\sum_{m=1}^M \alpha_m}}, \tag{8}$$

where the weights are generated based on the evaluated values as $\alpha_m = \exp\left[ \tau \left( V_{\Gamma_m^*} - \max_{m \in M} V_{\Gamma_m^*} \right) \right]$, with $\tau$ being the temperature coefficient.

**Iteration.** For the default setting, the above process are iterated $K$ times to derive the final action distribution. If the early-stopping heuristic is applied, after each iteration, we check whether the action optimization has converged by evaluating the KL divergence between the current and previous action distributions, $\mathbb{D}_{KL}(\mathcal{N}^{(k+1)} \| \mathcal{N}^{(k)}) < \eta$, where $\eta$ is a small threshold.

The detailed hyperparameters used in the model-based planner are summarized in Table 1.

## A.2   LOW-PASS ACTION SMOOTHING

In real-world robotics, high-frequency changes in control inputs can cause severe mechanical impacts, accelerating wear and reducing execution stability. Therefore, many studies in reinforcement learning and motion planning incorporate action-smoothing constraints, such as adding penalties on differences between consecutive actions during policy updates (Aractingi et al., 2023; Christmann et al., 2024; Wang et al., 2025), introducing regularization in policy networks (Chen et al., 2024b; Song et al., 2025), or filtering noise in trajectory optimization (Pinneri et al., 2021; Vlahov et al., 2024; Kicki, 2025), to reduce jitter and improve executability. Inspired by these methods, we apply frequency-domain low-pass filtering directly to the sampled action noise in our planner, explicitly suppressing the high-frequency components of the actions.

Specifically, during action sampling, we first sample noise from a standard normal distribution, apply low-pass filtering, and then add the filtered noise to the action mean to generate candidate action sequences. We use a Butterworth filter with $o_{\text{LBF}} = 1$, whose transfer function and amplitude–frequency response are given by:

$$H(s) = \frac{2\pi f_c}{s + 2\pi f_c}, \quad |H(f)| = \frac{f_c}{\sqrt{f^2 + f_c^2}}, \tag{9}$$

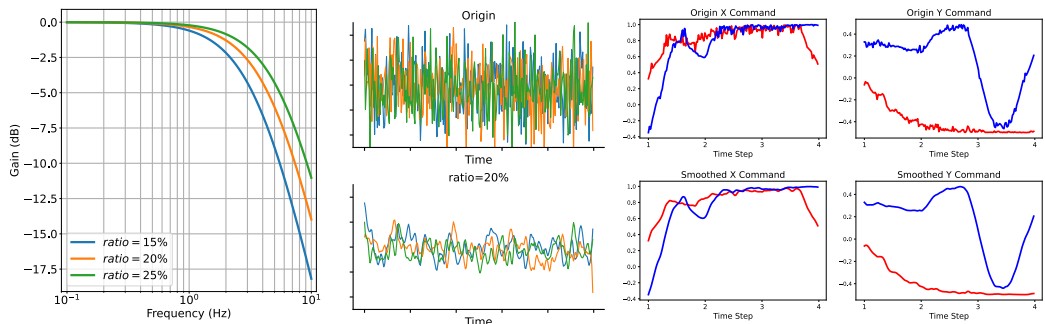

Figure 10: Visualization of the low-pass filter. (*Left*): Amplitude–frequency response of the low-pass filter. (*Middle*): Filtering effects on random signals at 20% cutoff ratios, the different colors represent different action dimensions. (*Right*): Effects of low-pass filtering on control commands in `PushBox`, with different colors representing different agents.

where $f_c$ is the low-pass cutoff frequency. As show in Figure 10, the amplitude–frequency response shows that the high-frequency components are exponentially attenuated (approaching linear decay in logarithmic coordinates). The corresponding discrete-time difference equation, obtained via bilinear transformation, is

$$y[t] = \frac{1-\beta}{2}(x[t] + x[t-1]) - \beta y[t-1], \quad \beta = \frac{1 - \tan\left(\pi f_c/f_s\right)}{1 + \tan\left(\pi f_c/f_s\right)}, \tag{10}$$

where $f_s$ is the sampling frequency, i.e., the frequency of the control signal.

## A.3 SERIAL-BLOCKING-FREE EXECUTION

A common concern regarding the sequential paradigm is that the $i$-th agent may need to wait for the $(i-1)$-th agent to finish planning, potentially causing inference time to grow linearly with the number of agents (Wen et al., 2022; Hu et al., 2025). This issue, referred as *serial blocking*, does not arise in SeqWM due to the use of the communication cache.

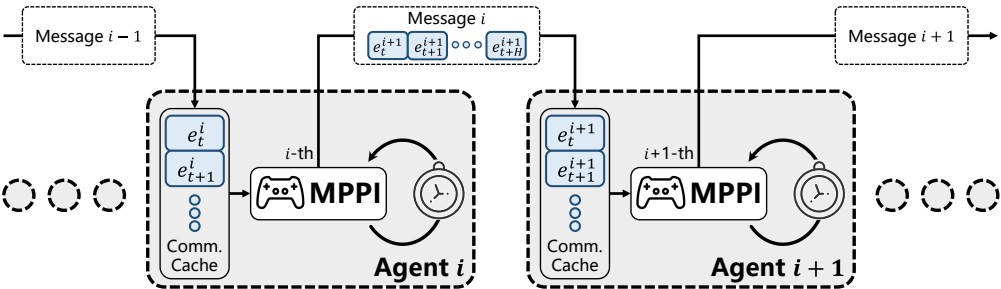

Figure 11: Sequential execution without serial blocking enabled by the communication cache.

As shown in Figure 11, the sequential paradigm specifies only the update order of the communication caches, rather than the execution order of each agent's planner. Each agent runs MPPI-based planner at a fixed control frequency and simply reads the latest available multi-step trajectory from the cache, without waiting for any other agent to complete planning. Under this design, the per-step inference latency satisfies

$$T_{\text{step}} \approx \max_i T_{\text{MPPI}}^{(i)} + T_{\text{comm}}, \tag{11}$$

meaning that SeqWM achieves $\mathcal{O}(1)$ decision latency with respect to the number of agents rather than exhibiting linear degradation.

---

**Algorithm 1** Model Training

---

**Input:** replay buffer $\mathcal{B}$, parameterized networks $\theta_E, \theta_D, \theta_R, \theta_Q$, and $\psi$ for encoder, dynamics predictor, reward predictor, critic, and actor, respectively;
**for** episode $= 1, 2, 3, \ldots$, **do**
    **for** step $t = 1, 2, 3, \ldots$ **do**
        Get real data $([o_t^i]_{i=1:n}, [a_t^i]_{i=1:n}, r_t, [o_{t+1}^i]_{i=1:n})$ by interacting with the environment
        Add transition into buffer: $\mathcal{B} = \mathcal{B} \cup ([o_t^i]_{i=1:n}, [a_t^i]_{i=1:n}, r_t, [o_{t+1}^i]_{i=1:n})$
    **end for**
    **for** epoch $= 1, 2, 3, \ldots$, **do**
        Sample trajectories from $\mathcal{B}$
        Update $\theta_E, \theta_D, \theta_R, \theta_Q$ by minimizing Eq. (3)
        Update $\psi$ by minimizing Eq. (4).
    **end for**
**end for**

---

**Algorithm 2** Model Planning

---

**Input:** learned parameters $\theta_E, \theta_D, \theta_R, \theta_Q, \psi$, hyperparameters $H, K, \tau, N_p, M, N_a$, initial distribution;
**for** step $t = 1, 2, 3, \ldots$ **do**
    **for** agent $i = 1, 2, \ldots, n$ **do**
        Get environment observation $o_t^i$ and encode it to latent space: $z_t^i = E^i(o_t^i)$
        **if** $i > 1$ **then**
            Retrieve the message from the previous agent $e_t^i = \bigoplus_{j<i} \left( \hat{z}_t^j, a_t^j \right)$
        **else**
            set $e_t^i = \emptyset$
        **end if**
        **for** iteration $= 1, 2, 3, \ldots, K_p$ **do**
            Sample $N_a$ actions $a_{t:t+H}^i \sim \mathcal{N} \left( \mu_{t:t+H}^i, (\sigma_{t:t+H}^i)^2 I \right)$
            Sample $N_p$ actions from actor $\hat{a}_h^i \sim \pi^{i,\text{Act}}(\cdot|o_h^i, e_h^i), h = t : t + H$
            Get predictions by world model rollouts, $\Gamma = \{(\hat{z}_h^i, a_h^i, \hat{r}_h^i)\}_{h=t:t+H}$
            Evaluate the trajectories by Eq. (7) and select top-$M$ elite action sequences
            Update action distribution following Eq. (8)
        **end for**
    **end for**
**end for**

---

# B IMPLEMENTATION DETAILS

## B.1 PSEUDOCODE

## B.2 HYPERPARAMETERS

We summarize the hyperparameters used in SeqWM in Table 1 and Table 2.

Table 1: The Notations and Values of hyperparameters in the planner.

| Hyperparameters | Notations | Value | Hyperparameters | Notations | Value |
|---|---|---|---|---|---|
| rollout horizon | $H$ | 3 | sampling actions | $N_p$ | 512 |
| planning iterations | $K$ | 6 | elites | $M$ | 64 |
| temperature | $\tau$ | 0.5 | actor samples | $N_a$ | 24 |

Table 2: The hyperparameters used in the world model.

| Hyperparameters | Value | Hyperparameters | Value | Hyperparameters | Value |
|---|---|---|---|---|---|
| batch size | 1000 | buffer size | 1e6 | dynamics coef | 20 |
| encoder lr scale | 0.3 | entropy coef | 1e-4 | lr | 5e-4 |
| n-step return | 20 | num bins | 101 | q coef | 0.1 |
| reward coef | 0.1 | step $\rho$ | 0.5 | | |

# C  ADDITIONAL EXPERIMENTS

## C.1  ADDITIONAL COMPARISONS

We report additional comparison results on other tasks to complement Figure 3.

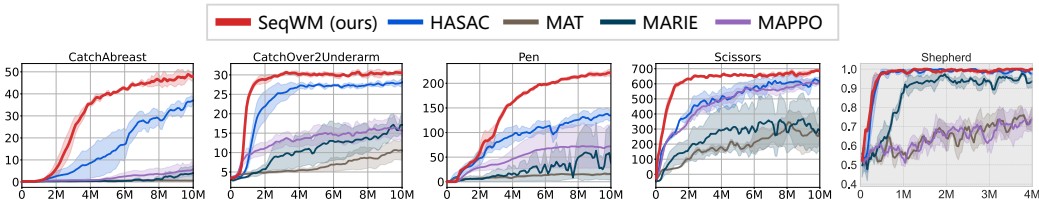

Figure 12: Comparison results on other tasks.

## C.2  INFERENCE TIME

**Inference Time Cost.** We report the per-step execution time of SeqWM on `BottleCap` using a single RTX A6000 GPU on the left side of Figure 13. The execution time increases almost linearly with the rollout horizon $H$ and the number of planner iterations $K$, which is consistent with the design of SeqWM. With the default settings, SeqWM achieves a per-step execution time of 12.8 ms, making it suitable for most real-time robotic tasks.

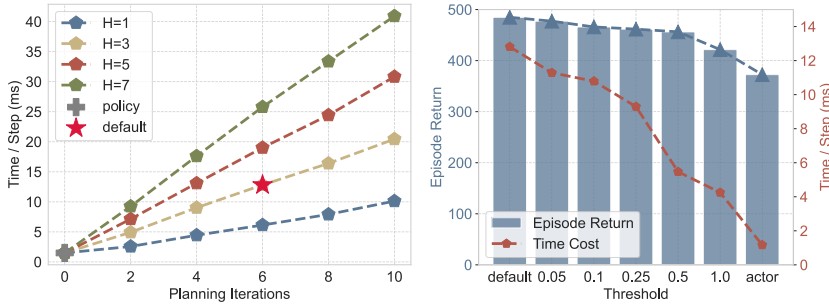

Figure 13: The per-step execution time of SeqWM. (*Left*): Time cost under different rollout horizons $H$ and planner iterations $K$. (*Right*): Time cost and performance with and without early-stopping heuristic.

**Early-Stopping Heuristic.** To further enhance the efficiency of SeqWM, we introduce an early-stopping heuristic in Section 4.2 that terminates iterations when the change in the action distribution is not significant. The KL divergence is used as a measure of distribution change, and the execution time and performance under different thresholds on `BottleCap` are shown on the right side of Figure 13. When the threshold is set to 0.5, SeqWM reduces the execution time by approximately 57.3% while incurring only about 5.9% performance loss.

## C.3  SIM-TO-REAL DEPLOYMENT

We implement all three Multi-Quad tasks in an 8m × 5m indoor space. Each task involves two Unitree Go2-W quadruped robots. The room is equipped with eight Mars cameras, and real-time localization of robots and objects is provided by the NOKOV 3D motion capture system.

- **PushBox**: We use a cardboard box of 1.2m × 1.2m × 0.5m and approximately 6 kg in mass. The box is sufficiently large that a single robot cannot independently control its movement direction, making cooperation essential. The static and kinetic friction coefficients between the box and the ground are both approximately 0.5.

- **Gate**: A 1m-wide doorway is set up. As shown in Figure 7 (b)-A, the two robots cannot pass through side-by-side, requiring coordinated navigation.

- **Shepherd**: A DJI EP robot acts as the guided agent (sheep). It is equipped with an omnidirectional chassis to simulate sheep behavior: it moves away from the nearest herding robot and its speed is inversely proportional to the distance to that robot.

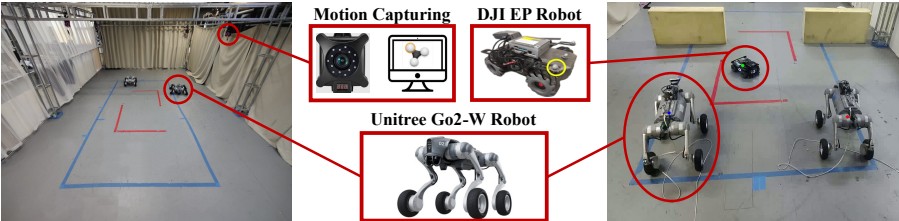

Figure 14: Real-world setups.

We employ the following strategies to enhance the generalization capability of SeqWM and facilitate sim-to-real transfer:

- **Observation transformation**: Positions of other robots are transformed from the global frame into the ego-centric frame of the current robot, reducing observation complexity and improving policy generalization.

- **Domain randomization**: Taking PushBox as an example, we randomize the initial positions/orientations of both robots and the box, the position and distance of the target, and the friction coefficient between the box and floor to improve robustness to environmental variations.

- **Sensor and actuation perturbations**: Random noise is added to sensor readings, and small delays with noise are introduced into control commands to emulate real-world sensing errors and actuation inaccuracies.

