# OpenReview forum: "Empowering Multi-Robot Cooperation via Sequential World Models"
_ICLR.cc/2026/Conference — ICLR 2026 Poster_

### Official Review · Reviewer_iNPz · 2025-10-29

**Soundness:** 3
**Presentation:** 2
**Contribution:** 2
**Rating:** 2
**Confidence:** 4

**Summary:**

This work proposes *SeqWM*, a framework for multi-robot cooperative learning. It first trains a Dreamer-style world model that predicts future rewards and observations, and then uses MPPI to derive actions based on the learned model. To facilitate multi-robot communication and cooperation, during the world model training stage, the $i+1$-th model receives the predictions from the first $i$ models. The proposed framework demonstrates its effectiveness across various tasks, both in simulation and in real-world experiments.

**Strengths:**

- Various experiments demonstrate the effectiveness of the proposed framework in both simulation and real-world settings. The experimental results show that the proposed method outperforms several baseline approaches.
- The authors provide extensive visualizations to illustrate the performance of their method, along with accompanying code and videos for more intuitive understanding.

**Weaknesses:**

- The presentation still has significant room for improvement. For example, the authors state that the world model predicts the “environment observation,” defined as the agent’s perception of the environment. However, it remains unclear what these observations actually are — joint angles, absolute positions, or visual inputs? Such important details are not explicitly mentioned in the paper.
- The discussion of related work is limited. While the authors review prior studies in model-based reinforcement learning and the sequential paradigm, they overlook a substantial body of work in multi-agent learning and multi-robot cooperation.
- The ablation studies are insufficient. The authors conduct ablation experiments on various communication function designs within the world model (Figure 9) and evaluate performance on the BottleCap task. However, other critical design choices, such as those in MPPI or world-model learning, remain unexplored. A key weakness is the lack of justification for why a world model is necessary for this task—see the detailed question below for further discussion.

**Questions:**

- A critical question for the proposed framework is: why are a world model and planning necessary in this context? From the demonstration videos, it appears that one major issue with baseline methods is that the robots lack awareness of each other’s positions(xy coordinates in world), leading to coordination failures. However, since motion capture data is available, one could simply provide each robot with the positions of the others and train them to complete the tasks. I would encourage the authors to explore these simpler baseline approaches and compare their performance with the proposed framework.
- It is also unclear why MAPPO performs poorly on certain tasks, such as PushBox. According to the Multi-Quad paper, MAPPO indeed shows weak performance on similar tasks, consistent with the authors’ findings here. Could the authors provide insights or analysis explaining why MAPPO struggles in these scenarios?

---

> ### Author Response · Authors · 2025-11-19
> **Authors' Rebuttal - (1/2)**
>
> `Responses`
> ---
> > **Q1: the authors state that the world model predicts the “environment observation,” and what information is included in the environment observations? (Weakness.1)**
>
> **A1:** Thank you for the careful review.
>
> 1. **Clarification of Misinterpretation:** We clarify that the world model in SeqWM does **not** predict raw environment observations. We think the ambiguity may come from the first paragraph of Section 4.1.
>
>     > This perspective reformulates multi-agent joint dynamics as a sequence modeling problem: given the token sequence $[(o^1\_{t}, a^1\_{t}), \dots, (o^n\_{t}, a^n\_{t})]$, the **model** generates the next-step outcomes $[(o^1\_{t+1}, r^1\_{t+1}), \dots, (o^n\_{t+1}, r^n\_{t+1})]$, ...
>
>     Here “the model” refers to a *conceptual joint dynamics model* for theoretical exposition, not the *learnable world model*. We have revised this to **“the dynamics generates”**  (Section 4.1, L152, P3).
>
> 2. **Latent-Space World Model:** As stated in next paragraph, as well as defined in Eq. (2), the learnable world model predicts the **latent** next-state, $\hat{z}\_{t+1}=D(z\_t, a\_t, e\_t)$, aligned with the encoded target $z\_{t+1} = E(o\_{t+1})$, rather than reconstructing $o_{t+1}$ directly. Reconstruction losses often overfit to irrelevant details [1], particularly in high-dimensional robotic tasks [2]. Following modern embodied world-model practice [3], SeqWM adopts a compact latent formulation. Ablations supporting this choice are provided in **A3.2** and **(Appendix C.9, L1591, P30).**
>
> 3. **Explicit Observation Definitions:** To avoid further confusion, we explicitly list the observation spaces, which keep consistent with prior work [4, 5]:
>
>    + **Bi-DexHands (e.g., Handover):** hand DoF states; fingertip pose/velocity/force/torque; hand base pose; last action; object pose/velocity; goal pose.
>
>    + **Multi-Quad (e.g., PushBox):** ego pose/orientation; **all other robots’ poses/orientations** (in ego frame); box center pose (ego frame).
>
> ---
> > **Q2: Why are a world model and planning necessary in this context? Are simpler baselines with explicit position sharing sufficient? (Questions.1)**
>
> **A2:** Thank you for the thoughtful question.
>
> 1. **Factual Clarification:** We do believe there is a factual misunderstanding that we would like to clarify. **All baselines already receive exactly the same observation space as SeqWM, including the full positions and orientations of all other robots.**
>
>    SeqWM uses no privileged information, and thus, the “simpler baseline” suggested by the reviewer is already implemented. Despite this, these baselines, including MAPPO, still fail on strongly coupled tasks such as PushBox and Gate. **Thus, the performance gap cannot be attributed to missing positional information.**
>
> 2. **Why “Seeing Others” Is Insufficient?** These tasks are highly coupled: each robot’s action instantly alters the global dynamics and the observations of all teammates. Knowing where others are at time $t$ does not reveal what they will do at time $t, t+1,..., t+H$, which is required for synchronized pushing and force alignment. A detailed analysis about why MAPPO fails are provided in **A4** and **(Appendix C.3.1, L1386, P26)**.
>
> 3. **Why World Models and Planning Are Necessary?** One of the motivations behind SeqWM is a simple, intuitive yet powerful insight:
>
>     > Cooperation becomes far easier when an agent agents know their partners’ future intentions, not just current states.
>
>     SeqWM enables this via:
>     1. **World model** for multi-step latent prediction;
>     2. **MPPI planner** for trajectory rollout;
>     3. **Sequential intention sharing** for exchanging predicted plans (intentions).
>
>     This mechanism allows each agent to align its behavior with future actions of others, enabling stable cooperation that baselines cannot achieve.
>
> 4. **Supporting Ablations.** To support the above claim, we added two additional ablations:
>    + DecWM: removes sequential communication; each agent predicts alone
>    + SeqFree: removes the world model; only one-step feature exchange
>
>    |SR(↑)|SeqWM|DecWM|SeqFree|
>    |-|-|-|-|
>    |Gate (2M)|**0.986**|0.647|0.245|
>    |Gate (4M)|**1.000**|0.980|0.995|
>    |PushBox (2M)|**0.993**|0.178|0.000|
>    |PushBox (4M)|**0.996**|0.892|0.000|
>
>    Key findings:
>    + SeqWM > DecWM >> SeqFree
>    + SeqFree fails, confirming that “more information” alone cannot induce cooperation.
>    + Only SeqWM, with multi-step intention sharing, achieves robust cooperation.
>     **For more details, please refer to (Section 5.5, L482, P9).**
>
> 5. **Takeaway.** SeqWM’s advantage does not come from extra information, and the baselines already have all positional data. The advantage arises from predicting and sharing future intentions, enabling anticipatory, coordinated multi-step behaviors that baselines cannot realize.

---

> ### Author Response · Authors · 2025-11-19
> **Authors' Rebuttal - (2/2)**
>
> ---
> > **Q3: Ablations on critical design choices in MPPI and world-model learning are missing. (Weakness.3)**
>
> **A3:** We provide more fine-grained ablations on planning and world-model design.
>
> 1. **Critic-Based Bootstrapping:** The critic in Eq. (5) supplies terminal values, reducing reliance on long horizons. Without it, the planner must use much larger horizon $H$. We compare two variants:
>    + Reward-only: no bootstrap
>    + Reward+Critic bootstrapped return
>
>    | Horizon | Reward+Critic | Reward-Only |
>    |-|-|-|
>    | | BottleCap (Return ↑) | |
>    | 3 | **480.4 ± 2.8** | 382.1 ± 9.6 |
>    | 5 | **483.1 ± 2.7** | 394.7 ± 7.9 |
>    | 7 | **482.9 ± 2.8** | 393.0 ± 6.7 |
>    | | Gate (Episode Length ↓)| |
>    | 3 | **135.9 ± 7.6** | 193.3 ± 25.7 |
>    | 5 | **134.1 ± 7.6** | 154.4 ± 24.6 |
>    | 7 | **134.0 ± 7.8** | 144.9 ± 21.4 |
>
>    Reward+Critic consistently outperforms Reward-only, especially at small $H$, confirming that value bootstrapping is crucial for stable and efficient planning. **For more details, please refer to (Appendix C.8, L1566, P30)**
>
> 2. **World Model Training Loss:** SeqWM trains entirely in latent space. To test reconstruction, we add a decoder and replace latent consistency with MSE reconstruction.We compare:
>
>    | Task | Latent Consistency | Observation Reconstruction |
>    |-|-|-|
>    | **BottleCap** (Return ↑) | **480.4 ± 2.8** | 238.2 ± 5.9 |
>    | **Gate** (Episode Length ↓) | **135.9 ± 7.6** | 355.6 ± 42.4 |
>
>     Reconstruction significantly degrades performance, showing that forcing full observation encoding harms compact, decision-centric latent representations. **For more details, please refer to (Appendix C.9, L1591, P30)**
>
> ---
> > **Q4: Why does MAPPO fail on PushBox? (Questions.2)**
>
> **A4:**
>
> + **Behavior Visualization:** As shown in (Figure 25, L1416, P27), MAPPO agents consistently display aimless random motion. Although they occasionally contact the box, they fail to produce consistent, aligned force, unable to learn even basic cooperation.
> + **Why MAPPO Fails?** PushBox is strongly coupled and highly non-stationary: the box moves only when robots push simultaneously and in similar directions. Each robot’s motion changes box dynamics and thus teammates’ observations, destabilizing MAPPO and leading to collapse, consistent with prior findings [5].
> + **Why SeqWM Succeeds?** SeqWM provides predicted multi-step teammate trajectories, allowing agents to align push direction and timing. Sequential intention sharing stabilizes these predictions despite dynamic coupling, enabling coordinated force generation.
>
> **For more details, please refer to (Appendix C.3.1, L1386, P26) and (Demo. 3, [here](https://sites.google.com/view/seqwm-marl/rebuttal)).**
>
> ---
> > **Q5: The discussion of related work on multi-agent learning and multi-robot cooperation is insufficient. (Weakness.2)**
>
> **A5:** Thank you for the constructive comment. We have expanded the discussion  (Section 2, L104, P2, revised paper).
>
> `Summary`
> ---
> Thank you for your valuable constructive feedback. We appreciate your recognition of the **performance** and **real-world validation** of our work, and we also acknowledge the positive affirmations from R1 (wPa1), R2 (hYcd), and R3 (VPtz) on:
>
> + **Novelty and Contribution:** (**R1:** *the method is an original combination*), (**R2:** *the central idea is novel and effective*), (**R3:** *has some value to the community*).
>
> + **Performance and Effectiveness:** (**R1:** *considerably improves sample efficiency*, *produce solutions to tasks that baselines cannot solve*), (**R2:** *good performance and sample efficiency*), (**R3:** *strong empirical results*).
>
> + **Real-World Validation:** (**R1:** *empirically validated w.r.t. real-world deployment*), (**R2:** *a strong contribution is the real-world demonstration*), (**R3:** *deployment to the real world further strengthened the result*).
>
> + **4. Clarity and Presentation:** (**R1:** *clear*, *core ideas are easily understandable*, *contribution is properly contextualized*), (**R2:** *well-organized, clearly written, and easy to follow*, *motivation and contributions are articulated very clearly*), (**R3:** *code, videos, and visualizations improve intuitive understanding.*).
>
> We believe the current review, rating or confidence may reflect some avoidable divergent readings, and we hope our responses address your concerns.We welcome any further suggestions and discussion. **If our response have resolved your concerns, we would sincerely appreciate your consideration for raising the score.**
>
> `References`
> ---
>
> [1] Learning latent dynamic robust representations for world models, ICML, 2024
>
> [2] A review of learning-based dynamics models for robotic manipulation, Science Robotics, 2025
>
> [3] A Comprehensive Survey on World Models for Embodied AI, arXiv, 2025
>
> [4] Towards human-level bimanual dexterous manipulation, IEEE T-PAMI, 2024
>
> [5] Mqe: Unleashing the power of interaction with multi-agent quadruped environment, IROS, 2024

---

> ### Author Response · Authors · 2025-11-24
> **Looking forward to further discussions!**
>
> Dear Reviewer iNPz,
>
> Thank you again for your thoughtful comments. We truly appreciate the time and care you have already put into the review.
>
> We have carefully addressed each of your questions and provided additional clarifications and new experiments in response to your concerns. If there are any points that remain unclear, we would be more than happy to further elaborate or provide additional analysis.
>
> Your feedback is extremely valuable to us, and we sincerely welcome any further thoughts or suggestions you may have.
>
> Best regards,
>
> Authors

---

### Official Review · Reviewer_VPtz · 2025-10-30

**Soundness:** 3
**Presentation:** 2
**Contribution:** 3
**Rating:** 6
**Confidence:** 3

**Summary:**

This paper proposes a model-based RL algorithm for multi-agent cooperative problems. The algorithm formulates the world model and the decision making as sequential. This design reduces the complexity of modeling the joint dynamics. The world model is used in a planner to get actions. Experiments conducted in simulated bidexhand and multi-quadruped tasks show that the sequential paradigm leads to more coordinated behaviors. The algorithm also achieves better sample efficiency compared to model-free and non-sequential model-based baselines.

**Strengths:**

1. The sequential paradigm is a reasonable design choice to reduce the complexity of a joint world model over multiple agents. Although modeling multi-agent problems as sequential is not new (also studied in one of the baseline methods), applying it to the learning of a dynamic model still has some value to the community.

2. The method achieves strong empirical results in simulation benchmark, achieving higher sample efficiency than all the baselines with a clear margin. The deployment to the real world further strengthened the result.

**Weaknesses:**

1. There might be some overclaim in contribution (3). Previous work has already demonstrated multi-quadruped pushing in the real world, such as [1].

[1] Learning Multi-Agent Loco-Manipulation for Long-Horizon Quadrupedal Pushing. ICRA 2025. https://arxiv.org/abs/2411.07104

2. Some figures are not self-contained. The captions in Figure 6 are too small to read and the message to convey here is not clear. The meaning of the y-axis in Figure 8 is not explained.

3. In Figure 3, all baselines achieve zero success in the Pushbox environment. Is there any explanation?

4. The ablation study on sequential sample generation seems a little incomplete. As we can see in Figure 8, the centralized architecture achieves even lower prediction errors than the sequential architecture. Does it mean that we should prefer centralized rather than the proposed sequential model?

**Questions:**

1. The sequential paradigm may consume longer inference time since the ${i+1}$-th agent depends on the inference result of the previous $i$ agents. How to overcome this issue when the number of agents is large?

2. Is the action execution of each agent asynchronized? If yes, do you take into account the asynchronous execution during training in simulation?

---

> ### Author Response · Authors · 2025-11-19
> **Authors' Rebuttal**
>
> `Responses`
> ---
>
> ---
> > **Q1: There might be some overclaim in contribution (3). Why do all baselines fail in PushBox? (Weakness.1 & 2)**
>
> **A1:**
> 1. We revised contribution (3) to emphasize general results rather than the specific task  (L73, P2)
>
> 2. Baselines fail because PushBox is strongly coupled and highly non-stationary:the box moves only under simultaneous, aligned forces. This makes coordination extremely unstable for pure reactivate MARL. Consistent with [1], MAT, MAPPO, and others also fail. Our visualizations (Fig. 25, L1416, P27) show that MAPPO agents wander, occasionally touching the box but failing to synchronize force or direction.
>
> 3. SeqWM succeeds by providing multi-step teammate predictions, enabling consistent direction and timing, which is key to generating coherent force.
>
> **For more details, please refer to (Appendix C.3.1, L1386, P26) and (Demo. 3, [here](https://sites.google.com/view/seqwm-marl/rebuttal)).**
>
> ---
> > **Q2: Does the centralized model’s lower prediction error in Fig. 8 imply it should be preferred over the proposed sequential architecture? (Weakness.4)**
>
> **A2:** Notably, **centralized model is generally impractical to deploy** in real systems due to full-state communication demands; it serves only as an upper bound. SeqWM remains practically deployable and stronger than decentralized prediction.
>
> To further validate, We added two ablations:
> + DecWM: removes sequential intention sharing (decentralized world model).
> + SeqFree: removes the world model entirely (only single-step messages).
>
> |SR(↑)|SeqWM|DecWM|SeqFree|
> |-|-|-|-|
> |Gate (2M)|**0.986**|0.647|0.245|
> |Gate (4M)|**1.000**|0.980|0.995|
> |PushBox (2M)|**0.993**|0.178|0.000|
> |PushBox (4M)|**0.996**|0.892|0.000|
>
> SeqWM performs best and DecWM cannot match SeqWM’s prediction quality or decision performance, consistent with the trends in Figure 8. **For more details about the ablation, please refer to (Section 5.5, L482, P9)**
>
> ---
> > **Q3: The sequential paradigm seems to consume longer inference time. How to overcome this issue when number of agents is large? (Questions.1)**
>
> **A3:** This concern applies only if each agent waited for its predecessor’s planning to finish. In SeqWM, this does not occur. With the communication cache, agents update the cache sequentially, but all planners run in parallel at a fixed control rate and simply read the latest available future trajectory, without blocking (Figure 20, L982, P19).
>
> The per-step latency $T_{\text{step}} \approx \max_i \, T_{\text{MPPI}}^{(i)} + T_{\text{comm}}$, i.e., $O(1)$ w.r.t. agent count, not linear.
>
> For larger populations, we also adopt a mean-field variant. SeqWM remains stable on MAgent with 45 and 134 agents.
>
> **For more details, please refer to (Appendix A.3, L958, P18), (Appendix C.3, L1280, P24) and (Demo. 1, [here](https://sites.google.com/view/seqwm-marl/rebuttal))**
>
> ---
> > **Q4: Is the action execution of each agent asynchronized, and if so, is this accounted for during training? (Questions.2)**
>
> **A4:** As noted in A3, actions are almost synchronous on our hardware, and we did not observe problematic asynchrony. Nevertheless, for better sim-to-real robustness, we inject small random actuation delays (<20 ms) during simulation training (Appendix C.7, L1564, P29), following standard practice [2].
>
> ---
> > **Q5: Small captions in Fig. 6 and the unexplained y-axis in Fig. 8. (Weakness.2)**
>
> **A5:** We updated Figure 6 with larger captions and corrected the y-axis labels of Figure 8 to Dynamics Prediction Error and Reward Prediction Error.
>
> ---
> `Summary`
> ---
>
> Thank you for your valuable and constructive feedback. We appreciate your recognition of the **performance** and **real-world validation** of our work, and we also acknowledge the positive affirmations from R1 (wPa1), R2 (hYcd), and R4 (iNPz) on:
> + **Novelty and Contribution:** (**R1:** *original combination*), (**R2:** *novel and effective*)
> + **Performance:** (**R1:** *considerably improves sample efficiency*), (**R2:** *good performance and sample efficiency*), (**R4:** *outperforms several baseline*)
> + **Real-World Validation:** (**R1:** *mpirically validated on real robots*), (**R2:** *a strong contribution is the real-world demonstration*)
> + **Clarity and Presentation:** (**R1:** *clear*, *easily understandable*, *well contextualized*), (**R2:** *well-organized, clearly written, and easy to follow*, *motivation and contributions are articulated very clearly*), (**R4:** *various experiments demonstrate the effectiveness*)
>
> We hope that our responses have fully addressed your concerns, and we welcome any further suggestions to improve this work. **If our clarifications have resolved your concerns, we would sincerely appreciate your consideration for raising the score.**
>
> ---
> `References`
> ---
>
> [1] Mqe: Unleashing the power of interaction with multi-agent quadruped environment, IROS, 2024
>
> [2] Dynamics Randomization Revisited: A Case Study for Quadrupedal Locomotion , ICRA, 2021

---

> > ### Comment · Reviewer_VPtz · 2025-11-26
> >
> > Thank you for your detailed response. My questions on the inference time and asynchrony are addressed. I will change the original score accordingly.

---

> > > ### Author Response · Authors · 2025-11-26
> > > **Thank You for Your Follow-Up**
> > >
> > > Dear Reviewer,
> > >
> > > Thank you very much for taking the time to read our response and for your follow-up comment. We truly appreciate your careful reassessment of the concerns, and we are glad that our clarifications were helpful.
> > >
> > > Your constructive feedback has been valuable in strengthening the quality of the paper. If you have any additional suggestions or further thoughts during the discussion phase, we would be more than happy to continue the conversation.
> > >
> > > Thank you again for your time and engagement in the review process.
> > >
> > > Best regards,
> > >
> > > The Authors

---

### Official Review · Reviewer_hYcd · 2025-10-30

**Soundness:** 4
**Presentation:** 4
**Contribution:** 3
**Rating:** 8
**Confidence:** 3

**Summary:**

This paper successfully integrates the sequential modeling paradigm into model-based multi-agent reinforcement learning and demonstrates both cooperative efficiency and real-world applicability. Addressing the challenging problem of multi-robot cooperation, it introduces a novel Sequential World Model framework. The proposed method not only outperforms existing state-of-the-art approaches in performance and sample efficiency but also achieves real-world deployment on physical robots, demonstrating its ability to handle complex cooperative tasks.

**Strengths:**

1. The central idea of reformulating joint dynamics modeling as an autoregressive sequential modeling problem is both novel and effective. The sequential design significantly reduces communication and modeling overhead, while enabling complex cooperative behaviors with multi-robots.
2. The paper is well-organized, clearly written, and easy to follow. The motivation and contributions are articulated very clearly.
3. The method achieves good performance and sample efficiency compared to all baselines. One of the strong contributions is the real-world demonstration. It successfully executed complex cooperative behaviors (e.g., pushing, yielding, shepherding) on physical robots. This reviewer believes that this strongly validates the practicality and robustness of the proposed approach.

**Weaknesses:**

1. The related work is well written, and it covers well model-based RL and the current sequential paradigm in multi-agent systems. It identifies the gap between centralized and decentralized approaches. However, this reviewer feels that deeper comparative analysis with closely related frameworks such as MARIE, CoDreamer, and SeqComm would clarify the unique technical contribution of SeqWM beyond general sequential coordination. Including reflections on potential limitations, such as dependence on agent ordering and communication latency, would also add balance and critical depth.
2. Although extensive and well-structured, this reviewer feels that the ablation experiments are not fully sufficient to validate the claimed contributions. First, there is no direct ablation isolating the effect of the sequential structure on emergent cooperative behaviors such as predictive adaptation, temporal alignment, and role division. It is unclear whether these behaviors arise from the sequential design, the MPPI planner, or improved world model accuracy. Second, it appears that the ablations are performed primarily on BottleCap, not on multi-robot coordination tasks such as Multi-Quad or PushBox, limiting generalization to core cooperative scenarios. Third, other design elements, including the planner, early stopping, and communication cache, are under-analyzed. While briefly mentioned in the appendix, the main text lacks quantitative justification of their contributions. For example, there is no comparison showing whether cooperation emerges even without the sequential structure, such as a “Planner-only” baseline.
3. In sequential planning, with N agents, the Nth agent must wait for all preceding agents to finish planning, which may introduce latency. While the paper demonstrates scalability, it does not analyze how this latency affects responsiveness in larger systems. A discussion on the theoretical convergence and stability of the sequential updates would strengthen the paper.
4. The proposed method’s reliance on a fixed agent order raises questions about sensitivity. There is no in-depth analysis of how agent ordering affects performance or how accumulated prediction errors propagate over long horizons. (1) How is the order determined, manually or learned? (2) What happens if the optimal order changes during execution (for example, in the Shepherd task when the leading robot should switch)? The paper mentions random masking during training to improve robustness, which is a good idea, but it remains unclear whether a fixed or randomized order is used during testing and how this choice affects robustness.
5. While the sequential scheme reduces communication frequency, transmitting entire predicted trajectories as messages may increase communication bandwidth. As N grows, the message size received by later agents could scale linearly, potentially becoming a limitation or constraint.

**Questions:**

1. If this reviewer understands correctly, SeqWM assumes a leader–follower (predecessor–successor) structure where agents act in a predefined order (e.g., Agent 1 -> Agent 2 -> Agent 3). Would this design limit its applicability to tasks that require true simultaneity or are inherently symmetric without a natural ordering? For instance, how would SeqWM handle scenarios where two robots must react simultaneously to catch a fast-moving, unpredictable object, or where multiple robots must lift a shared payload at the exact same time to maintain stability?
2. How computationally heavy is the proposed planner? how does the runtime scale as the number of agents increases, and what are the practical limits for real-time deployment in physical multi-robot systems?
3. Based on Table 2, it appears that each agent performs approximately K=6 iterations. If that is correct, does the last agent need to wait for all preceding agents to complete their iterations? For example, with five agents, what is the observed planning delay (in ms) for the final agent? At what point (in terms of the number of agents) does sequential latency become a bottleneck for real-time control?

---

> ### Author Response · Authors · 2025-11-19
> **Authors' Rebuttal - (1/2)**
>
> `Responses`
> ---
>
> > **Q1.1: Deeper comparative analysis with closely related frameworks such as MARIE, CoDreamer, and SeqComm would helpful. (Weakness.1)**
>
> **A1.1:** We have expanded the comparative analysis to clarify the position of SeqWM.
>
> + MARIE requires communication at every prediction step for aggregation, whereas SeqWM uses autoregressive agent-wise prediction, avoiding step-wise synchronization.
>
> + CoDreamer builds multi-agent world models using GNNs and a two-level communication mechanism, which limits real-world deployment. SeqWM instead adopts independent and agent-wise models with a sequential structure for scalable deployment.
>
> + SeqComm applies the sequential paradigm but lacks multi-step predictive planning. SeqWM extends this paradigm by integrating sequential world modeling + planning, enabling explicit intention sharing.
>
>
> > **Q1.2: The paper lacks potential limitations, such as dependence on agent ordering and communication latency. (Weakness.1)**
>
> **A1.2:** We have added detailed discussions of both issues in Appendix A3 and A4, and included a dedicated limitations paragraph in Conclusion (L517, P10).
>
> ---
> > **Q2.1: Missing ablation isolating the sequential paradigm’s contribution to cooperation. (Weakness.2)**
>
> **A2.1**: Thank you for the careful assessment. We have expanded the ablation studies to directly address all raised concerns.
>
> 1. **Sequential Intention Sharing:** To validate SeqWM’s central mechanism, predicting and sharing multi-step future intentions, we compare:
>    + DecWM: no sequential intention sharing
>    + SeqFree: no world model (only single-step messages)
>
>    |SR(↑)|SeqWM|DecWM|SeqFree|
>    |-|-|-|-|
>    |Gate (2M)|**0.986**|0.647|0.245|
>    |Gate (4M)|**1.000**|0.980|0.995|
>    |PushBox (2M)|**0.993**|0.178|0.000|
>    |PushBox (4M)|**0.996**|0.892|0.000|
>
>    SeqWM > DecWM >> SeqFree, showing that both multi-step prediction and sequential sharing play essential roles . **For more details, please refer to (Section 5.5, L482)**
>
> 2. **Critic-Based Bootstrapping:** The critic in Eq. (5) supplies terminal values, reducing reliance on large horizons. We compare two variants:
>    + Reward-only: no bootstrap
>    + Reward+Critic bootstrapped return
>
>    |Horizon|Reward+Critic|Reward-Only |
>    |-|-|-|
>    ||BottleCap (Return ↑)||
>    |3|**480.4 ± 2.8**|382.1±9.6|
>    |5|**483.1 ± 2.7**|394.7±7.9|
>    |7|**482.9 ± 2.8**|393.0±6.7|
>    ||Gate (Episode Length ↓)||
>    |3|**135.9 ± 7.6**|193.3±25.7|
>    |5|**134.1 ± 7.6**|154.4±24.6|
>    |7|**134.0 ± 7.8**|144.9±21.4|
>
>    The results show that bootstrapping yields consistently better performance, especially at small $H$. **For more details, please refer to (Appendix C.8, L1566, P30)**
>
> 3. **World Model Training Loss:** Replacing latent consistency with reconstruction (decoder+MSE) significantly degrades performance:
>
>    |Task|Latent Consistency|Observation Reconstruction|
>    |-|-|-|
>    |**BottleCap** (Return ↑)|**480.4 ± 2.8**|238.2 ± 5.9|
>    |**Gate** (Episode Length ↓)|**135.9 ± 7.6**|355.6 ± 42.4|
>
>    This confirms that forcing full observation encoding harms decision-centric latent structure. **For more details, please refer to (Appendix C.9, L1591, P30)**
>
>
> > **Q2.2: Ablations only on BottleCap, not on multi-robot tasks, limit generalization. (Weakness.2)**
>
> **A2.2:** We extend the experiments in Figure 8 to Gate and PushBox, with results reported in (Figure 15, L1176, P22).
>
>
> ---
> > **Q3: How does sequential planning latency scale with the number of agents, and does it become a bottleneck for real-time control? (Weakness.3 & Questions.3)**
>
> **A3:** This concern applies only if each agent waited for its predecessor’s planning output. In SeqWM, this does not occur. With the communication cache, agents update the cache sequentially, but all planners run in parallel at a fixed control rate and simply read the latest available future trajectory, without blocking (Figure 20, L982, P19).
>
> The per-step latency $T_{\text{step}} \approx \max_i \, T_{\text{MPPI}}^{(i)} + T_{\text{comm}}$, i.e., $O(1)$ w.r.t. agent count, not linear.
>
> For larger populations, we also adopt a mean-field communication variant. SeqWM remains stable on MAgent with 45 and 134 agents.
>
> **For more details, please refer to (Appendix A.3, L958, P18), (Appendix C.3, L1280, P24) and (Demo. 1, [here](https://sites.google.com/view/seqwm-marl/rebuttal)).**

---

> ### Author Response · Authors · 2025-11-19
> **Authors' Rebuttal - (2/2)**
>
> ---
> > **Q4: How is the agent order determined, and how sensitive is SeqWM to this ordering? (Weakness.4)**
>
> **A4:**
> 1. **Sensitivity to ordering.** As noted in other sequential MARL literature (e.g., PMAT), the communication order can influence performance. SeqWM addresses this by randomly permuting the order during training (Section 4.1), forcing each agent-wise world model to be robust to different predecessor–successor configurations.
>
> 2. **How the order is determined.** At test time, we use a fixed but randomly sampled order. Approaches that learn an adaptive order (e.g., PMAT) require a centralized referee or full global access, which is incompatible with real-world deployment requirements.
>
> 3. **What if the “optimal” order changes during execution?** We agree this may lead to suboptimal results and could cause SeqWM to fall back toward DecWM-like performance. Thank you for the insightful question, and determining an optimal order under limited communication is an promising direction for future work.
>
> ---
> > **Q5: Does trajectory-level messaging create a bandwidth bottleneck as N increases? (Weakness.5)**
>
> **A5:**  We agree that current scheme increases communication bandwidth as $N$ grows. This can be mitigated by replacing the concat-based function with more compact alternatives such as an MLP or cross-attention module, which keep the message dimensionality stable. As shown in (Figure 9 (left), L480, P9), these reduced-bandwidth variants incur a modest performance drop but still outperform all model-free and model-based baselines. This suggests that SeqWM remains effective even under constrained communication budgets.
>
> ---
> > **Q6: Does the sequential structure limit SeqWM’s applicability to tasks requiring true simultaneity or symmetric coordination? (Questions.1)**
>
> **A6:** We believe that sequential intention sharing is an advantage rather than a limitation. This enables forward-looking synchronization: agents align their upcoming behaviors before simultaneous execution occurs.
>
> For example, if one robot predicts an interception at $t+3$, its predicted future trajectory is passed to other agents, allowing them to adjust their motions and arrive at the target point at the same timestep. This forward-looking synchronization is fundamentally different from leader–follower actuation and does not impose action delays.
>
> To verify this, we introduce a Gate-Sync task, SeqWM learns precise synchrony, robots match speed, align side-by-side, and cross simultaneously.
>
> **For more details, please refer to (Appendix C.4, L1424, P27) and (Demo. 2, [here](https://sites.google.com/view/seqwm-marl/rebuttal))**
>
> ---
> > **Q7: How heavy is the planner computationally, and how does its runtime scale with the number of agents? (Questions.2)**
>
> **A7:**
> 1. SeqWM inference consists of four parts:
>    + **Sampling**: generating action candidates via the actor;
>    + **Rollout:** unrolling latent dynamics for $H$ steps;
>    + **Evaluation:** applying the critic to terminal latent states;
>    + **Optimization:** selecting elite samples and updating the action distribution.
>
>     |Module|Sampling|Dyna.|Reward|Eval.|Opt.|
>     |-|-|-|-|-|-|
>     |Time (%)|19.6%|10.2%|11.1%|1.5%|57.6%|
>
>     This indicates that the bottleneck lies in optimization rather than model inference, complementing the early-stopping analysis and clarifying SeqWM’s computational applicability. **For more details, please refer to (Appendix C.6, L1485)**
>
> 2. As noted in A3, its runtime is $\mathcal{O}(1)$ w.r.t. the number of agents
>
> ---
> `Summary`
> ---
>
> Thank you for the valuable and constructive feedback. We appreciate your recognition of the **novelty**, **performance**, **real-world validation** and **clarity** of our work, and we also acknowledge the positive affirmations from R1 (wPa1), R3 (VPtz) and R4 (iNPz) on:
>
> + **Novelty and Contribution:** (**R1:** *the method is an original combination*), (**R3:** *has some value to the community*)
> + **Performance and Effectiveness:** (**R1:** *considerably improves sample efficiency*, *produce solutions to tasks that baselines cannot solve*), (**R3:** *strong empirical results*), (**R4:** *outperforms several baseline*)
> + **Real-World Validation:** (**R1:** *empirically validated w.r.t. real-world deployment*), (**R3:** *deployment to the real world further strengthened the result*).
> + **Clarity and Presentation:** (**R1:** *the manuscript is clear*, *core ideas are easily understandable*, *contribution is properly contextualized*), (**R3:** *code, videos, and visualizations improve intuitive understanding.*), (**R4:** *various experiments demonstrate the effectiveness in both simulation and real-world*)
>
> We hope these clarifications address your concerns, and we would be grateful for your positive consideration. If our clarifications have resolved your concerns, we would sincerely appreciate your consideration for keeping or further raising the score.

---

### Official Review · Reviewer_wPa1 · 2025-10-31

**Soundness:** 4
**Presentation:** 3
**Contribution:** 3
**Rating:** 8
**Confidence:** 3

**Summary:**

The paper presents sequential world models (SeqWM), a novel multi-agent reinforcement learning (MARL) method that incorporates the sequential multi-agent paradigm into model-based MARL. In SeqWM agents have sequential, independent autoregressive world models, each conditioned on the predecessor, avoiding reliance on centralized stable communication channels. SeqWM is compared with both model-free and model-based baselines on various tasks from two simulation benchmarks, in which SeqWM improves both over final performances and sample efficiency. Additional experiments (ablations, scalability, real-world deployment) further solidify the relevance of SeqWM.

**Strengths:**

Quality:
- reproducibility is supported by publicly available code and hyperparameters
- the presented method is compared to relevant model-free and model-based baselines
- the method is investigated on multiple simulated domains and in the real-world

Significance:
- results on the simulated domains suggest that SeqWM not only considerably improves sample efficiency, but can also produce solutions to tasks that baselines cannot solve
- the role of the core idea (sequential paradigm in model-based MARL) is empirically demonstrated with an ablation
- the method is empirically validated w.r.t. scalability and real-world deployment

Originality:
- the method is an original combination of well-established ideas, namely the sequential multi-agent paradigm and world models

Clarity:
- overall, the manuscript is clear and the layout of the content is predictable
- core ideas are easily understandable
- the contribution is properly contextualized both in the introductory and related work sections

**Weaknesses:**

Quality:
- although the ablations presented are meaningful, more fine-grained ablations would better support the various contributions to the improved performance of SeqWM w.r.t. baselines; however, this would require substantial additional work which might be out of the scope of this work

Significance:
- while computational costs for the planner early-stopping is properly investigated, computational costs of the other compontents is not reported. Discussing this either in the main text or appendix would help framing the applicability of SeqWM
- while SeqWM is validated on challenging and varied robotics-based domains, it is unclear whether its benefits also apply to different kind of domains (e.g. virtual multi-agent environments like StarCraft); focus on robotics environments is nonetheless properly motivated and sufficiently significant as is

Clarity:
- while the presented method is clear, pseudocode is not present in the main text (but it is present in appendix B.1); moving it or just briefly referencing it in the main methodology section would make it easier to follow
- the y axis of figure 8 is confusing: it is reported as "accuracy", but the main text mentions that the decentralized setting is worse, even though its bars are higher (possibly a typo)
- appendix B.1 contains no text besides the pseudocode, which is confusing. Adding simple sentence in the main text of appendix B.1 referencing algorithms 1 and 2 would improve its readability

**Questions:**

- is the y axis of figure 8 reporting accuracy (more is better) or loss (less is better)?
- how general is SeqWM beyond robotics environments?
- is SeqWM applicable to environments with very high number of agents (e.g. swarms)? What are its limitations in these scenarios?
- how are compute costs distributed among the various components of SeqWM? What are the computational bottlenecks in the reported experiments?

---

> ### Author Response · Authors · 2025-11-19
> **Authors' Rebuttal**
>
> `Responses`
> ---
>
> ---
> > **Q1: More fine-grained ablations could better assess SeqWM’s contributions. (Weakness.Quality)**
>
> **A1:** Following your suggestions, we added 3 fine-grained ablations.
>
> 1. **Sequential Intention Sharing:** To validate SeqWM’s central mechanism, predicting and sharing multi-step future intentions, we compare:
>    + DecWM: no sequential intention sharing
>    + SeqFree: no world model (only single-step messages)
>
>    |SR(↑)|SeqWM|DecWM|SeqFree|
>    |-|-|-|-|
>    |Gate (2M)|**0.986**|0.647|0.245|
>    |Gate (4M)|**1.000**|0.980|0.995|
>    |PushBox (2M)|**0.993**|0.178|0.000|
>    |PushBox (4M)|**0.996**|0.892|0.000|
>
>    SeqWM performs best, showing that both multi-step prediction and sequential sharing play essential roles. **For more details, please refer to (Section 5.5, L482, P9)**
>
> 2. **Critic-Based Bootstrapping:** The critic in Eq. (5) supplies terminal values, reducing reliance on large horizons. We compare two variants:
>    + Reward-only: no bootstrap
>    + Reward+Critic bootstrapped return
>
>    |Horizon|Reward+Critic|Reward-Only |
>    |-|-|-|
>    ||BottleCap (Return ↑)||
>    |3|**480.4 ± 2.8**|382.1±9.6|
>    |5|**483.1 ± 2.7**|394.7±7.9|
>    |7|**482.9 ± 2.8**|393.0±6.7|
>    ||Gate (Episode Length ↓)||
>    |3|**135.9 ± 7.6**|193.3±25.7|
>    |5|**134.1 ± 7.6**|154.4±24.6|
>    |7|**134.0 ± 7.8**|144.9±21.4|
>
>    The results show that bootstrapping yields consistently better performance, especially at small $H$. **For more details, please refer to (Appendix C.8, L1566, P30)**
>
> 3. **World Model Training Loss:** Replacing latent consistency with reconstruction (decoder+MSE) significantly degrades performance:
>
>    |Task|Latent Consistency|Observation Reconstruction|
>    |-|-|-|
>    |**BottleCap** (Return ↑)|**480.4 ± 2.8**|238.2 ± 5.9|
>    |**Gate** (Episode Length ↓)|**135.9 ± 7.6**|355.6 ± 42.4|
>
>    This confirms that forcing full observation encoding harms decision-centric latent structure. **For more details, please refer to (Appendix C.9, L1591, P30)**
>
> ---
> > **Q2: How general is SeqWM beyond robotics environments? (Weakness.Significance.1 & Questions.2)**
>
> **A2:** We believe that SeqWM is not restricted to robotics; its core insight, treating world-model-predicted future trajectories as explicit intentions shared across agents, is agnostic to the underlying domain. To extend SeqWM to SMAC, we:
> 1. Use one-hot actions to transform a discrete action into a continuous one.
> 2. Replace the MPPI control with Cross Entropy Method (CEM).
>
> With these changes, SeqWM achieves 100% win rate on the SMAC '3m' scenario, indicating that sequential intention sharing generalizes beyond physical robots. We also conduct additional experiments on MAgent, please refer to A4.
>
> ---
> > **Q3: What are SeqWM’s compute distribution and main bottlenecks? (Weakness.Significance.2 & Questions.4)**
>
> **A3:** SeqWM inference consists of four parts:
> + **Sampling**: generating action candidates via the actor;
> + **Rollout:** unrolling latent dynamics for $H$ steps;
> + **Evaluation:** applying the critic to terminal latent states;
> + **Optimization:** selecting elite samples and updating the action distribution.
>
> |Module|Sampling|Dyna.|Reward|Eval.|Opt.|
> |-|-|-|-|-|-|
> |Time (%)|19.6%|10.2%|11.1%|1.5%|57.6%|
>
> This indicates that the bottleneck lies in optimization rather than model inference, complementing the early-stopping analysis and clarifying SeqWM’s computational applicability. **For more details, please refer to (Appendix C.6, L1485, P28)**
>
> ---
> > **Q4: Is SeqWM applicable to environments with very high number of agents (e.g. swarms)? (Questions.3)**
>
> **A4:** For swarm-scale settings, we adopt a mean-field variant to keep communication and rollout costs tractable. A new swarm-level experiments on MAgent with 45 agents and 134 agents show stable learning and efficient inference. **For more details, please refer to (Appendix C.2, L1283, P24) and (Demo. 1, [here](https://sites.google.com/view/seqwm-marl/rebuttal)).**
>
> ---
> > **Q5: The authors should better reference the pseudocode and clarify the confusing y-axis labeling in Figure 8. (Weakness.Clarity & Questions.1)**
>
> **A5:** We added explanatory text in Appendix B.1 to better reference the pseudocode. For Figure 8, the y-axis indeed denotes prediction error, not accuracy; the caption has been corrected.
>
> ---
> `Summary`
> ---
> Thank you for the valuable feedback. We appreciate your recognition of the **novelty**, **performance**, **real-world validation** and **clarity** of our work. We hope these clarifications address your concerns, and we would be grateful for your positive consideration. If our clarifications have resolved your concerns, we would sincerely appreciate your consideration for keeping or further raising the score.

---

### Author Response · Authors · 2025-11-24
**Authors' Global Response (Before Discussion Freeze)**

This message has been entirely superseded by the **Authors' Final Summary**.

---

### Meta-Review · Area_Chair_yn7H · 2026-01-07

**Summary:**

This paper was reviewed by four reviewers, with 3 positive recommendations and 1 negative recommendations. The major concerns are limited discussion of related work and lack of ablation studies. After carefully reading the comments and responses, the AC believes the major concerns have been well addressed.

**Reviewer Concerns:**

The major concerns about limited discussion of related work and lack of ablation studies have been addressed.

**Reviewer Scores:**

The last reviewer may change the rating score to positive

---

### Decision · Program_Chairs · 2026-01-26

Accept (Poster)